# Conscious observational behavior in recognizing landmarks in facial expressions

**Kuangzhe Xu**[1]*, **Toshihiko Matsuka**[2]*

**1** Institute for Promotion of Higher Education, Hirosaki University, Aomori, Japan, **2** Department of Cognitive and Information Science, Chiba University, Chiba, Japan

* jokyotetsu@hirosaki-u.ac.jp (KX); matsuka@chiba-u.jp (TM)

**Data Availability Statement:** Our datasets, stan model and R code generated during this study are available on OSF at https://osf.io/bf94z/.

**Funding:** The author(s) received no specific funding for this work.

## Abstract

The present study investigated (1) how well humans can recognize facial expressions represented by a small set of landmarks, a commonly used technique in facial recognition in machine learning and (2) differences in conscious observational behaviors to recognized different types of expressions. Our video stimuli consisted of facial expression represented by 68 landmark points. Conscious observational behaviors were measured by movements of the mouse cursor where a small area around it was only visible to participants. We constructed Bayesian models to analyze how personality traits and observational behaviors influenced how participants recognized different facial expressions. We found that humans could recognize positive expressions with high accuracy, similar to machine learning, even when faces were represented by a small set of landmarks. Although humans fared better than machine learning, recognition of negative expressions was not as high as positives. Our results also showed that personality traits and conscious observational behaviors significantly influenced recognizing facial expressions. For example, people with high agreeableness could correctly recognize faces expressing happiness by observing several areas among faces without focusing on any specific part for very long. These results suggest a mechanism whereby personality traits lead to different conscious observational behaviors and recognitions of facial expressions are based on information obtained through those observational behaviors.

## Introduction

Despite controversy regarding facial expression classification, most researchers now apply the six types of expression classification (happiness, sadness, anger, disgust, surprise, and fear) proposed by Ekman et al. [1, 2]. These six types of facial features have been used in variety of studies (e.g., assessment of human expressive ability, understanding of others' emotions, and cross-cultural differences in facial expression recognition) [3]. The ability to recognize facial expressions is essential for many human social activities and has been used as an important method of understanding human social cognition. Many researchers have paid attention to the recognition of facial expressions because the universality of facial expressions has proven to be extremely effective in understanding the nature of humans and their relationships with society [4, 5].

**Competing interests:** The authors declare no conflict of interest.

Studies have confirmed that facial areas serve as clues for recognizing facial expressions. For example, Ito et al. reported a strong influence in the upper area of the face in recognizing anger, sadness, and surprise and a strong influence in the lower area of the face in recognizing fear and happiness [6]. Studies using video stimuli also showed that specific responses (eye movements and observational behaviors) were associated with recognizing different facial expressions [7–9].

As technologies evolved, research on facial expression recognition using machine learning techniques also flourished [10, 11]. Some results were surprisingly consistent with previous facial expression recognition studies using human participants: most clues were concentrated in the mouth and/or eyes, with little reference to other parts of the face (hair, ears, etc.). One possible reason for such results is the method of extracting facial features. Feature extraction using facial landmarks is now the major method for face recognition in machine learning [12]. This method focuses only on 2-dimensional facial features, represented by relatively small numbers of landmarks, and does not consider other information such as hair, ears, and depths of features. Nevertheless, this method was applied and showed high accuracy in some facial expression recognition [13].

In recent years, the relationship between facial features and observational behavior has attracted much attention [14, 15] and this trend is also applies to facial expression recognition. Studies using eye trackers showed that, for example, when looking at faces expressing happiness or disgust, the observers tended to look at the upper lip. On the other hand, when looking at faces expressing anger, fear, or sadness, they tended to look at the eyes [16]. Another study differentiating between a genuine smile and a presented smile showed that judging the genuineness of a smile based on a single static smile image alone is challenging [17]. In the same study, when genuine and presented smile faces were presented simultaneously, participants exhibited different observational behaviors and were able to discern whether the smile was genuine. Although some studies examining relationships between facial expression recognition and observational behavior showed slightly lower accuracies with video stimuli than static images [18, 19], observational behaviors differed among different facial expressions in both video and static images.

Accumulating bodies of knowledge revealed some essential aspects of the relationships between facial expression recognition and observational behavior, but several questions remain unanswered. First, facial expression recognition accuracies differ between humans and machine learning. Machine learning achieved similar levels of accuracy as humans for positive facial expressions, such as happiness and surprise. But, it failed to correctly recognize negative facial expressions, such as sadness and fear. That is, the same machine learning approach—using a limited set of facial landmarks—has led to similar levels of accuracy compared to humans in some expressions while worse in others. The main reason for this may be that while the machine learning approach generally uses very sparsely represented facial images using a limited number of landmarks, humans use richer information on faces. Almost no research has examined whether humans can accurately recognize facial expressions when the faces are sparsely represented by landmarks. Second, previous studies showed that individuals' observational behaviors strongly correlated with their personality traits, and this relationship remained apparent even when eye movements were manipulated [14]. The results indicate that there are two types of observational behavior: conscious (controlled) and unconscious (uncontrolled or impulsive). The eye movement data recorded by eye trackers may contain a mixture of these two types of observational behavior, and there is no study, to the best of our knowledge, that verified how conscious observational behavior differs when humans recognize facial expressions.

In this study, we created video facial stimuli represented by landmarks to examine the following two research questions: (1) How well humans can recognize facial expressions

represented by landmarks, a commonly used technique in facial recognition with machine learning; and (2) differences in conscious observational behaviors when recognizing various expressions. Note that we left out the effect of unconscious observational behaviors in our research questions, as we were only able to develop a method to collect data on conscious observational behaviors but not unconscious behaviors.

## Materials and methods

### Participants

Forty-one students from Hirosaki University participated in the experiment: 12 male and 29 female participants. Their mean age was 20.31 (SD = 1.368). All participants received a 1,000-yen gift card for participating in the experiment. The recruitment period was from October to December 2022. We have deleted all identifiable information about the participants, and the data are securely stored. All procedures and processes of this experiment were reviewed by Hirosaki University Faculty of Education Ethics Review Committee (receipt number 0007(2022)). Before the experiment began, participants received a brief explanation about the experiment and signed a consent form to participate.

### Stimuli

In this experiment, we used a facial expression video database developed by the National Institute of Advanced Industrial Science and Technology (AIST) of Japan [20]. The database includes 12 expression types with eight Asian actors (four males and four females). Among them, we selected six types of facial expression, namely happiness, sadness, anger, disgust, surprise, and fear. Each video lasts 2500 ms, including a 500 ms neutral face in the beginning then a 2000 ms emotional expression. Using these videos, we extracted 68 facial landmarks (x & y coordinates) using Dlib API for each frame. We then created animated gif images of those extracted facial landmarks. All animated gif images were configured to repeatedly play 10 times. The resolution of the animated gifs was $600 \times 800$ pixels. As shown in Fig 1, the facial landmarks occupied only about 30%—40% of the screen. Therefore, the actual size of the faces was approximately $200 \times 250$ pixels. There were a total of 48 (eight actors × six expression types) unique animated gif videos.

### Procedure

The experiment consisted of two main parts, each containing 48 sessions (videos). Before starting the main experiment, each participant had a practice session to become familiar with the tasks and stimuli. After the practice session, they were directed to the main parts.

In Part 1, participants were presented with the animated gif images of six emotional facial expressions and asked to choose which emotions each video exhibited. At the beginning of every session, we reminded participants about the task, and then participants started the task on their timing. In each session, one of the videos was presented randomly after a 0.5-s gazing symbol. In each session, participants were able to see a video as long as they wanted (but at most ten times), so participants were able to observe the video until they were ready to confirm their response. When participants clicked a mouse button, they were asked to choose one emotion type that the stimulus exhibited. Participants were asked to select one expression that they thought best fit the video and then to rate how confident they were in their decision on a 7-point scale. Once the decision was made, they moved on to the next session. After all sessions of Part 1 were completed, there was a 3–5 minute break.

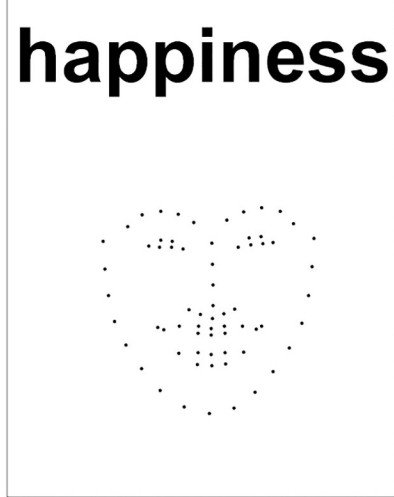
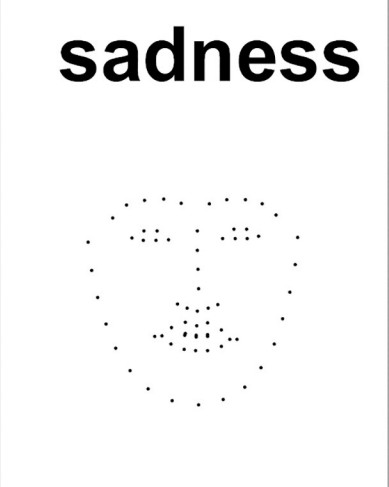
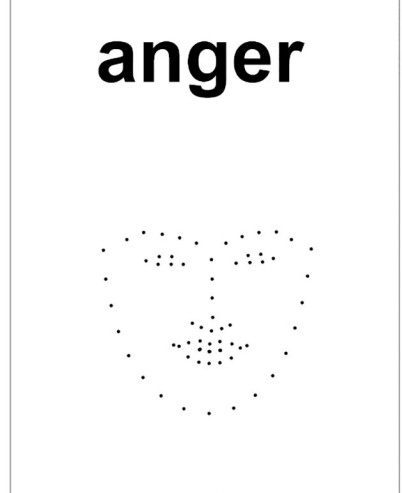
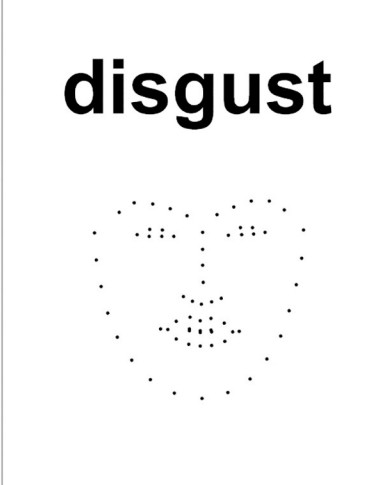
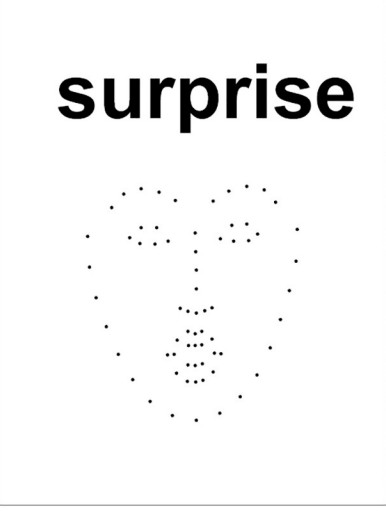
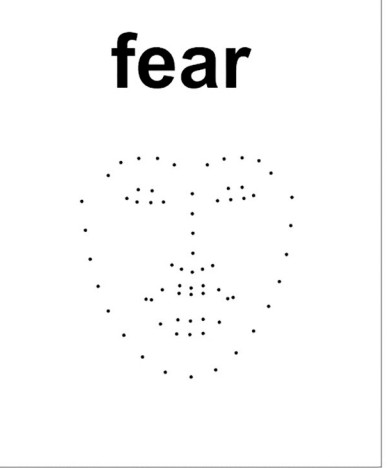

**Fig 1. Examples of one video frame for each of the six emotional expression types.**

The procedure in Part 2 was essentially the same as in that of Part 1. However, facial stimuli was masked so that only a portion around the mouse cursor was visible to participants (Fig 2). By moving the mouse cursor, participants could see the area they wished to observe. We consider this means of measuring areas that participants wish to look at reflect conscious observational behavior. The visible area was circular with a radius of 20 pixels, and then smoothed with a 20 pixels Gaussian filter to obtain a smooth edge. The size of the visible area was determined according so that roughly one-third of the face video was observable.

After completing the recognition task, participants were asked to complete the Japanese version of the Ten Item Personality Inventory (TIPI) to measure their five personality traits (i.e., Agreeableness, Conscientiousness, Extraversion, Neuroticism, and Openness) [21]. TIPI-J consists of 10 questions, in which each factor of the Big Five (Big Five personality traits are Extraversion, Agreeableness, Openness, Conscientiousness, and Neuroticism. In the

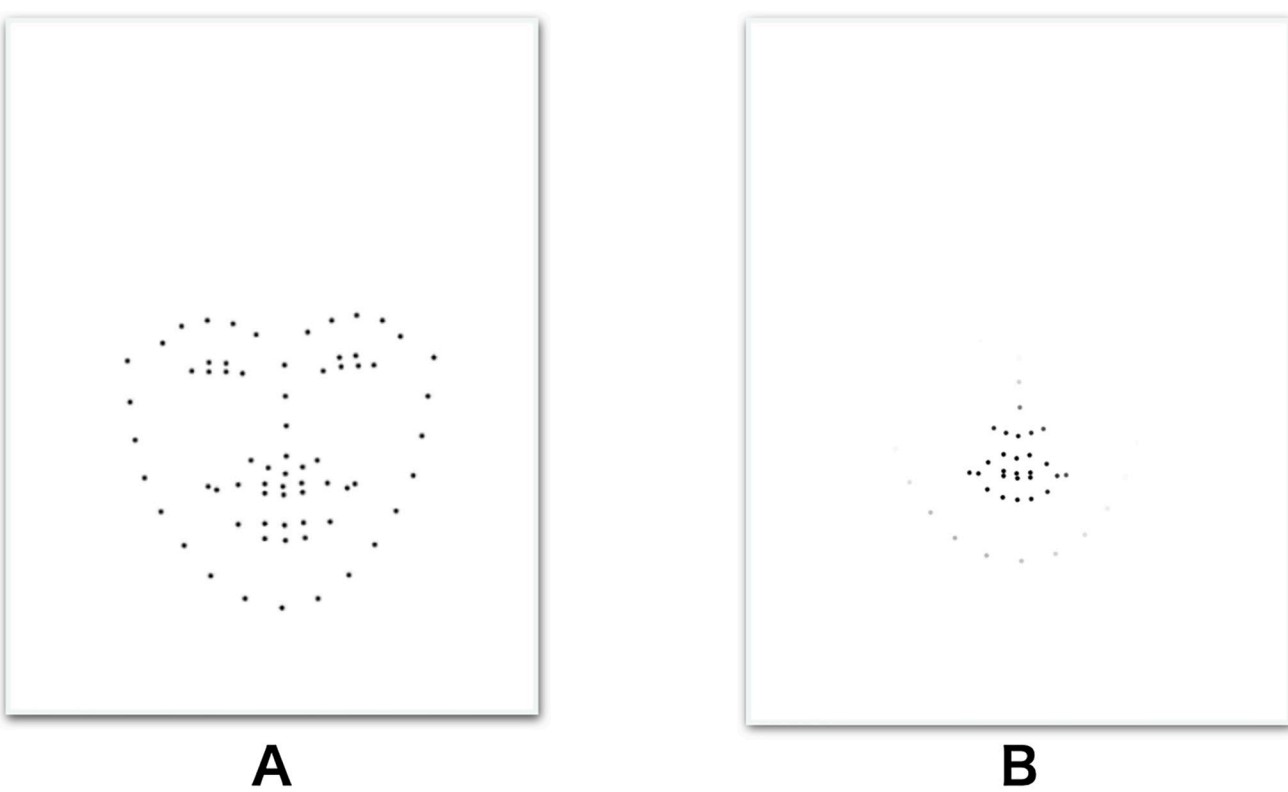

**Fig 2. Example of faces shown/visible to participants in Part 1 (A) and Part 2 (B).**

previous study [22], the Big Five was briefly described as follows: Extraversion includes sociability, activity, assertiveness, and positive emotionality. Agreeableness encompasses traits like altruism, tender-mindedness, trust, and modesty. Conscientiousness comprises traits of thinking before acting, delaying gratification, following norms and rules, and planning, organizing, and prioritizing tasks. Neuroticism is characterized by emotional stability, such as feeling anxious, nervous, sad, and tense. Openness reflects the breadth, depth, originality, and complexity of an individual's mental and experiential life.) was measured by two questions (positive and negative). For each question, respondents were asked to rate on a 7-point scale from "I strongly disagree" (1 point) to "I strongly agree" (7 points)." All experimental procedures were conducted on a personal computer, and we used PsychoPy to run and control the experiment.

## Data preprocessing

We preprocessed the recorded mouse movement data to analyze the relationships between conscious observational behavior and emotion recognition. Specifically, the coordinates of the cursor were converted into the coordinates on the computer screen by superimposing the location of the stimuli presented on the screen. We applied a Gaussian filter with a 10-pixel standard deviation to the converted data on the basis on the previous research [23]. Filtering was applied to all frames and all filtered data within a session was aggregated to create one weighted data for each session. Referring to previous studies [8], we divided facial stimuli into three areas, namely upper, middle, and lower parts (upper: middle: low = 1:0.8:0.6) as shown in Fig 3.

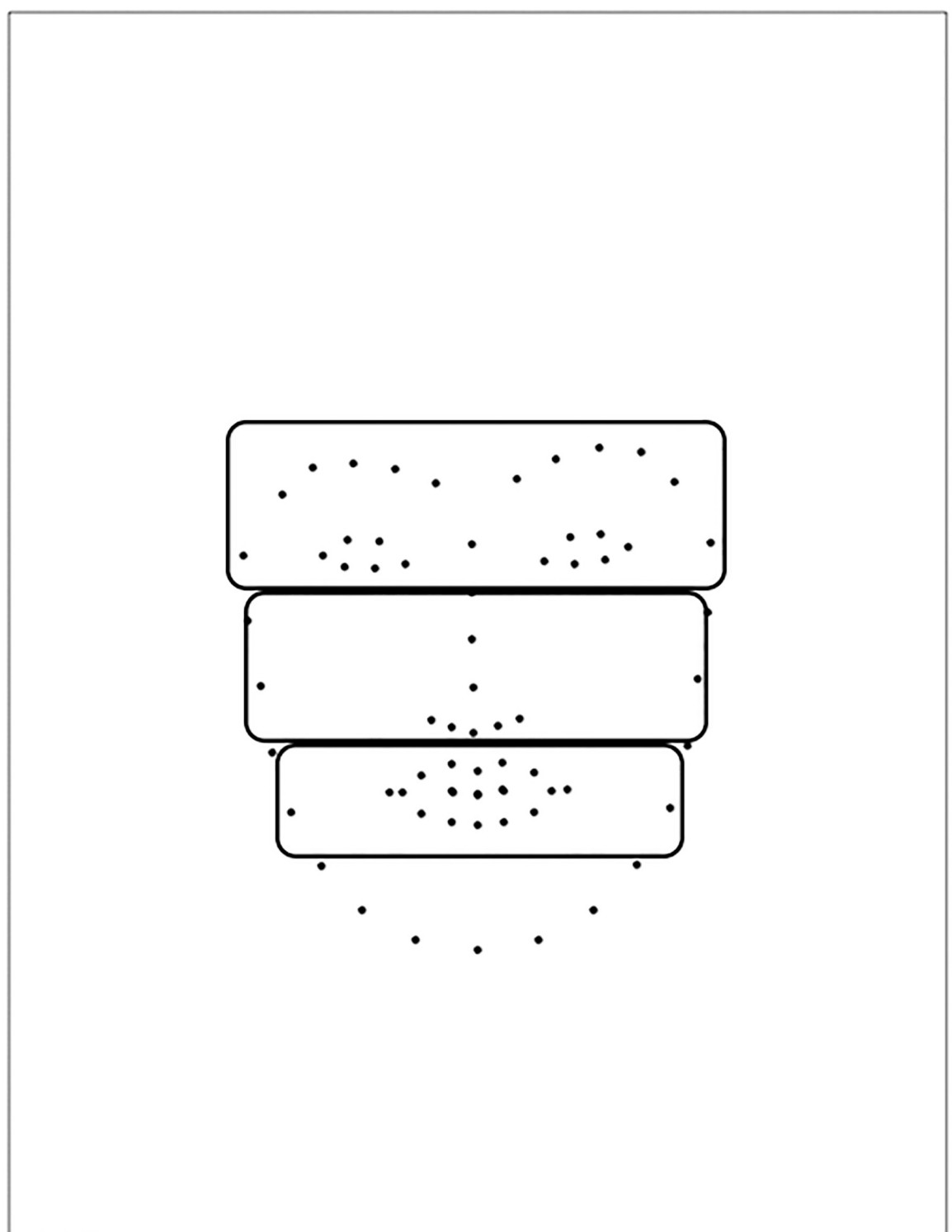

**Fig 3. Example of partitioning a face into three parts.**

## Analysis

We conducted three separate analyses for experimental data. Analysis A examined the relationship between participants' observational behavior and facial expression recognition. This analysis aimed to test whether specific observational behaviors were associated with specific facial expressions.

Analysis B examined the relationship between participants' personality traits and their choice of facial expressions. This analysis aimed to test whether people with different personality traits respond differently to specific facial expressions.

Analysis C examined the relationship between participants' personality traits and their observational behaviors. This analysis aimed to test whether personality traits affect observed behaviors when judging facial expressions.

In Analyses A and C, the data from only Part 2 were used in the model analysis. This is because the main role of Part 1 regarding to Analyses A and C was to allow participants to become familiar with emotional facial videos represented by landmarks and to obtain some ideas about which parts of faces to look at to differentiate different expressions when observational behaviors were restricted. Therefore, the data on Part 1 had no direct impact on the main of Analyses A and C and were excluded from the analyses.

In all analyses, we constructed a Bayesian hierarchical model to analyzed the data. Bayesian statistics enables reliable estimation and prediction by incorporating prior knowledge and deriving posterior distributions from the data. Particularly, as demonstrated in this study, Bayesian methods provide accurate parameter estimation even with limited data. Moreover, the use of prior distributions helps prevent overfitting in complex models. Leveraging these properties, we estimated observers' personality traits and observed behavioral tendencies in facial expression judgments using the Bayesian approach.

**Analysis A.** Since there were six alternative facial expressions, the choices or responses of expressions can be described by multinomial logistic distribution, and modeled the reasons using a hierarchical Bayesian model. The specific equations are as follows:

$$\prod_{1 \leq k \leq 6} \text{Bernoulli}(y_k \mid \text{logit}^{-1}(\mu_{ij})) \tag{1}$$

$$= \prod_{1 \leq k \leq 6} \begin{cases} \text{logit}^{-1}(\mu_{ij}) & \text{if } y_k = 1, \text{ and} \\ 1 - \text{logit}^{-1}(\mu_{ij}) & \text{if } y_k = 0. \end{cases} \tag{2}$$

$$\mu_{ij} = \alpha_{kg} + \sum_{1 \leq g \leq 3} G_{ijg} \cdot \beta_{kg} + r_{kg}^{subj} + r_{jg}^{stim} \tag{3}$$

$y_k$ is a variable indicating whether the participant made a choice for each of the expressions (happiness, sadness, anger, disgust, surprise, or fear). The objective variable of the constructed multinomial logistic was 1 if the participant made a choice and 0 if not.

Eq 3 shows the relationship between the variables ($\mu_{ij}$) in estimating the multinomial logistic and the observational behavior ($G_{ijg}$). $\alpha_{kp}$ represents the intercept, and $\beta_{kp}$ represents the fixed effect of each observational behavior. The $i$ and $j$ variables denote the participants and stimulus video indices, respectively. $G_{ijg}$ represents the observation behavior $g$(upper, middle, or low) of participant $i$. $r_{ig}^{subj}$ and $r_{jg}^{stim}$ represents the random effects of the participant and the stimulus video, respectively.

We used Rstan [24–26] for parameter estimations. The weakly informative prior was used for fixed effects (normal with $\mu = 0$, $sd = 10$) and random effects (gamma with $\alpha = 10$, $\beta = 10$). We used Rstan's default settings for MCMC (Markov chain Monte Carlo methods) sampling.

For each model, there were four chains, each of which had 1,000 warmup steps and 2,000 iterations. Thus, there were a total of 4,000 MCMC samples for each model.

To verify whether MCMC samplings had converged, we checked $\hat{R}$ values. $\hat{R}$ values for all coefficients were less than 1.1, a widely used criterion, and we considered that our MCMC sampling had converged.

None of the coefficients in the model were significantly related to judgments (i.e., 95% Highest Density Interval or HDI included 0). The HDI credible intervals are the Bayesian version of confidence intervals. Thus, the HDI specifies an interval that spans most of the distribution such that every point inside the interval has higher credibility than any point outside the interval.

**Analysis B.** As the objective variable remained about selections of facial expressions, a multinomial logistic model was constructed as in Analysis A. However, the explanatory variable was replaced with personality traits instead of observational behavior. The equation is as follows: $P_{ip}$ represents the personality trait $p$(big five) of participant $i$. $r_{ip}^{subj}$ and $r_{jp}^{stim}$ represent the random effects of the participant and the stimulus video, respectively. Model estimation was performed in Rstan, with the same weak prior information and each parameter to be estimated as in Analysis A. The HDI was also used to determine significant differences.

$$\mu_{ij} = \alpha_{kp} + \sum_{1 \leq p \leq 5} P_{ip} \cdot \beta_{kp} + r_{ip}^{subj} + r_{jp}^{stim} \tag{4}$$

**Analysis C.** As shown in Fig 4, the distribution of observational behavior may be best described by a mixed distribution with (1) zeros or non-zeros and (2) a continuum between values greater than zero and less than or equals to one.

Referring to previous studies [27, 28], we constructed a zero-inflated beta distribution that explains such a distribution well. The most important feature is its ability to explain complex situations using two distributions: the Bernoulli distribution, which estimates whether the distribution is zero or not, and the Beta distribution, which estimates a non-zero distribution. This feature can be used to predict the probability of whether or not a person has taken an

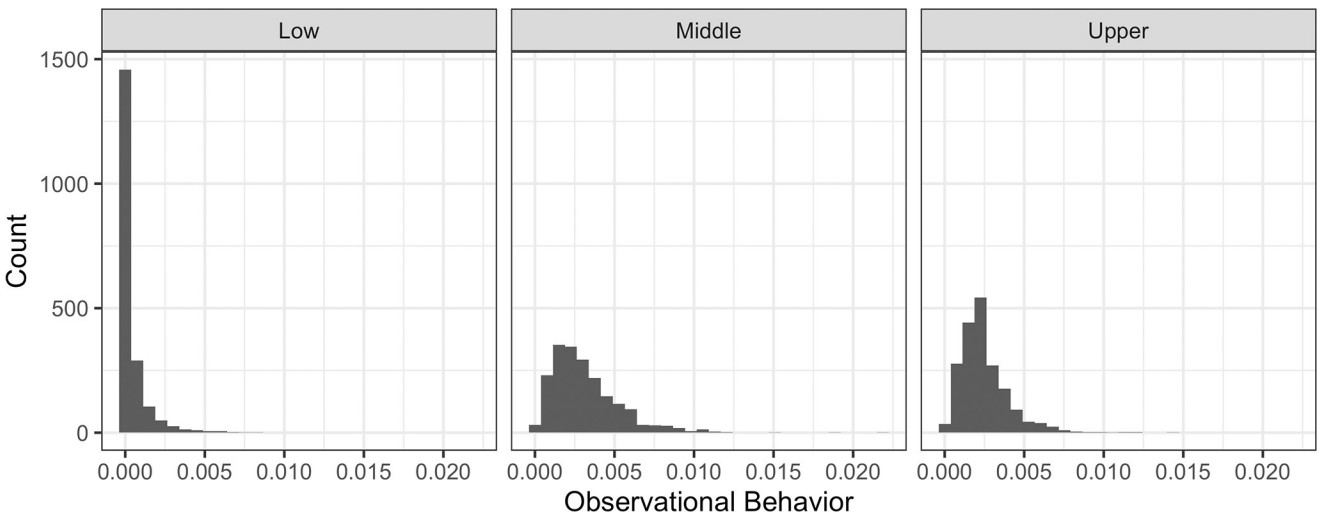

**Fig 4. Distributions of observational behaviors (accumulated weights).**

observational behavior from a specific area and by how much. The equations are given as follows:

$$G_{ijk} \sim \text{ZIB}(q_{ijk}, a_{ijk}, b_{ijk}) \qquad (5)$$

$$\text{ZIB}(G_{ijk}|q_{ijk}, a_{ijk}, b_{ijk}) =$$

$$\begin{cases} Bern(0|q_{ijk}) & (G_{ijk} = 0) \\ Bern(1|q_{ijk}) \times Beta(G_{ijk}|a_{ijk}, b_{ijk}) & (G_{ijk} > 0) \end{cases} \qquad (6)$$

$$q_{ijk} = \frac{1}{1 + \exp(-(\alpha_k^Z + \sum_{l=1}^{5} \beta_{kl}^Z p_{il} + r_{ik}^Z + r_{jk}^Z))} \qquad (7)$$

$$a_{ijk} = \phi \cdot \mu_{ijk} \qquad (8)$$

$$b_{ijk} = \phi(1 - \mu_{ijk}) \qquad (9)$$

$$\mu_{ijk} = \frac{1}{1 + \exp(-(\alpha_k^B + \sum_{l=1}^{5} \beta_{kl}^B p_{il} + r_{ik}^B + r_{jk}^B))} \qquad (10)$$

$G_{ijk}$ is the observational behavior, and $i$, $j$, and $k$ are the participants, stimulus videos, and indices of different observational behaviors (upper, middle, or low), respectively. $q_{ijk}$ is a parameter used to estimate the Bernoulli distribution, and its relationship to the participants' personality traits is expressed in Eq (7). $a_{ijk}$ and $b_{ijk}$ are the parameters for estimating the beta distribution and represent the relationship between the personality traits of the participants in Eqs (8) to (10). $P_{ik}$ represents the personality trait $l$(big five) of participant $i$. The prior distribution of fixed effects followed the normal distribution with a mean of 0 and a variance of 100. The prior distribution of random effects followed the gamma distribution ($\alpha = 10$, $\beta = 10$). The HDI was also used to determine significant differences.

## Results

### Result 1

In Part 1, we examined the relationship between personality traits and selections about expressions where there was no restriction on observations. Because we did not measure observational behaviors in Part 1, only Analysis B was conducted for Part 1. Fig 5 shows the estimated selection tendencies for each stimulus video's expression in the model, with the dashed line indicating the chance level. While positive expressions (happiness and surprise) tended to be recognized correctly, negative expressions were recognized not as high as positives. In particular, there was a tendency to mistakenly recognize fear as surprise or happiness. These results are consistent with previous studies using machine learning for expression recognition, but humans are superior to machine learning in accurately recognizing sadness.

Table 1 summarizes the significant correlations between participants' personality traits and selections of facial expressions. The results show that people with high Agreeableness tended to correctly recognize happiness. Those with high Extraversion tended to correctly recognize surprise. On the other hand, people with high Openness tended to misidentify anger as sadness and fear as disgust. Those with high Conscientiousness tended to misidentify surprise as anger. Additionally, people with high Neuroticism tended to avoid selecting fear as fear and

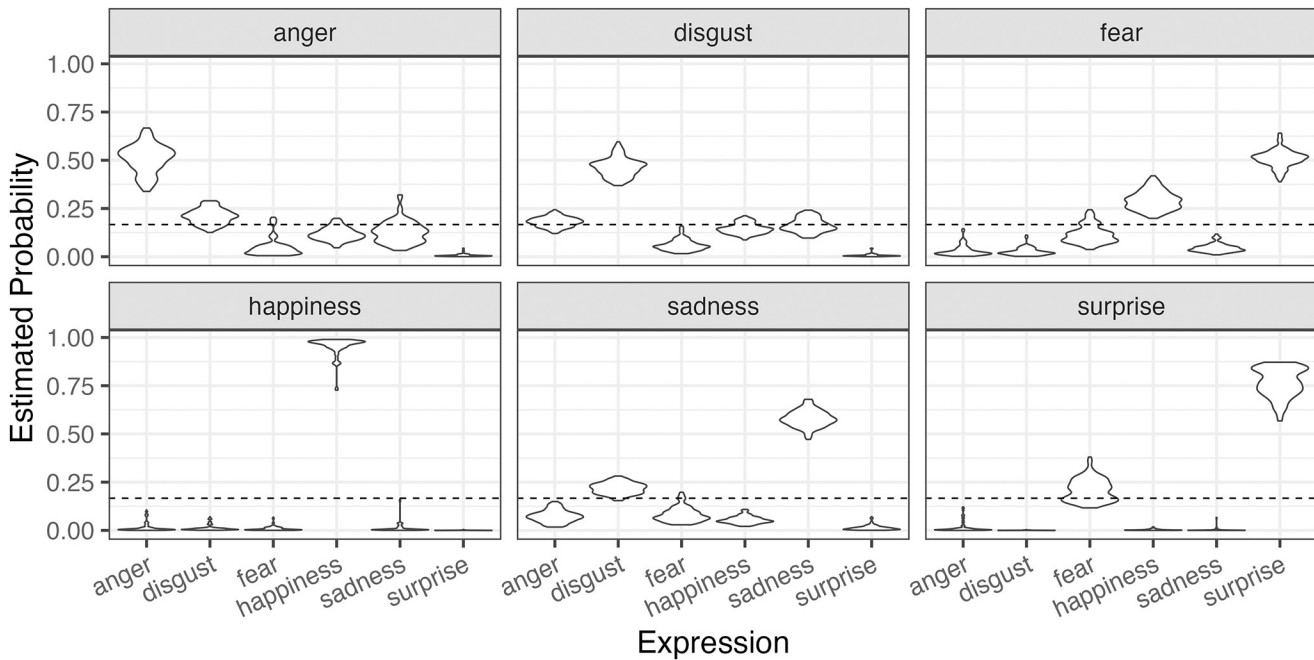

**Fig 5. Predictive posterior distributions of selecting each expression type for each condition (emotional expression) in Experiment Part 1.**

anger as fear. Therefore, in addition to the difficulty in recognizing facial expressions represented by landmarks, the influence of participants' personality traits cannot be ignored as a cause of inaccurate recognition of facial expressions.

Furthermore, to compare how changes in the five personality traits affect the probability of selecting each expression, we simulated the results using the model. Figs 6 to 9 show the

**Table 1. Significant personality traits for selecting and or not selecting each expression type for each condition in Experiment Part 1.**

| Expression | Response | Predictor | Mean | 95%HDI |
|---|---|---|---|---|
| Happiness | Happiness | Agree. | 0.672 | 0.264 ⌣ 1.066 |
| | Anger | Agree. | -1.147 | -1.852 ⌣ -0.474 |
| | Disgust | Agree. | -0.973 | -1.733 ⌣ -0.258 |
| | Sadness | Agree. | -0.939 | -1.652 ⌣ -0.252 |
| Sadness | Anger | Consc. | -0.393 | -0.750 ⌣ -0.023 |
| Surprise | Anger | Consc. | 0.931 | 0.123 ⌣ 1.736 |
| | | Extra. | -0.933 | -1.624 ⌣ -0.182 |
| | Surprise | Extra. | 0.277 | 0.066 ⌣ 0.481 |
| | Fear | Extra. | -0.223 | -0.438 ⌣ -0.003 |
| Anger | Sadness | Open. | 0.258 | 0.003 ⌣ 0.526 |
| | | Extra. | -0.362 | -0.640 ⌣ -0.115 |
| | Anger | Open. | -0.174 | -0.340 ⌣ -0.007 |
| | Fear | Neuro. | -0.534 | -0.861 ⌣ -0.195 |
| Fear | Disgust | Open. | 0.591 | 0.033 ⌣ 1.183 |
| | Anger | Extra. | -0.585 | -1.096 ⌣ -0.127 |
| | Fear | Neuro. | -0.239 | -0.461 ⌣ -0.010 |

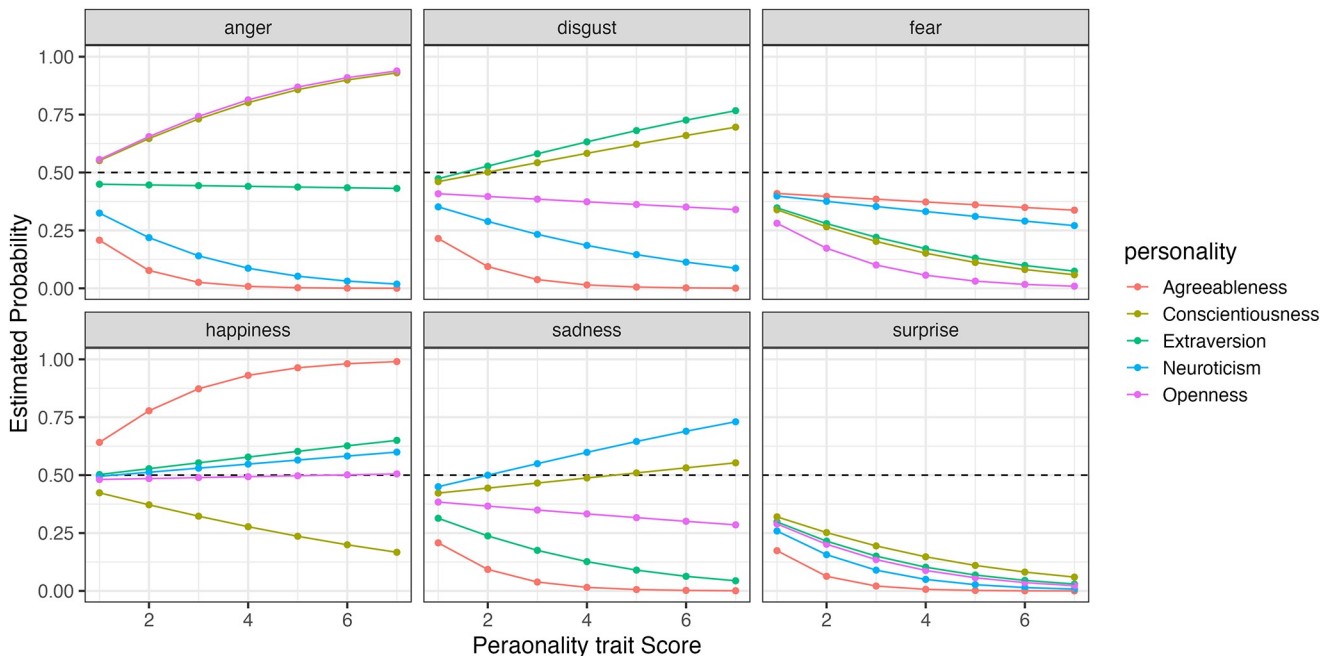

**Fig 6. The probability of selecting each expression as the five personality traits changed when faces expressing happiness were presented in Experiment Part 1.**

example of the probability of selecting each expression as the five personality traits change when evaluating happiness, sadness, surprise and fear (the results of all analyses including data are available at OSF https://osf.io/bf94z/). We examined the changes in the selection probability of each expression when one of the five personality traits was varied from 1 to 7, while the

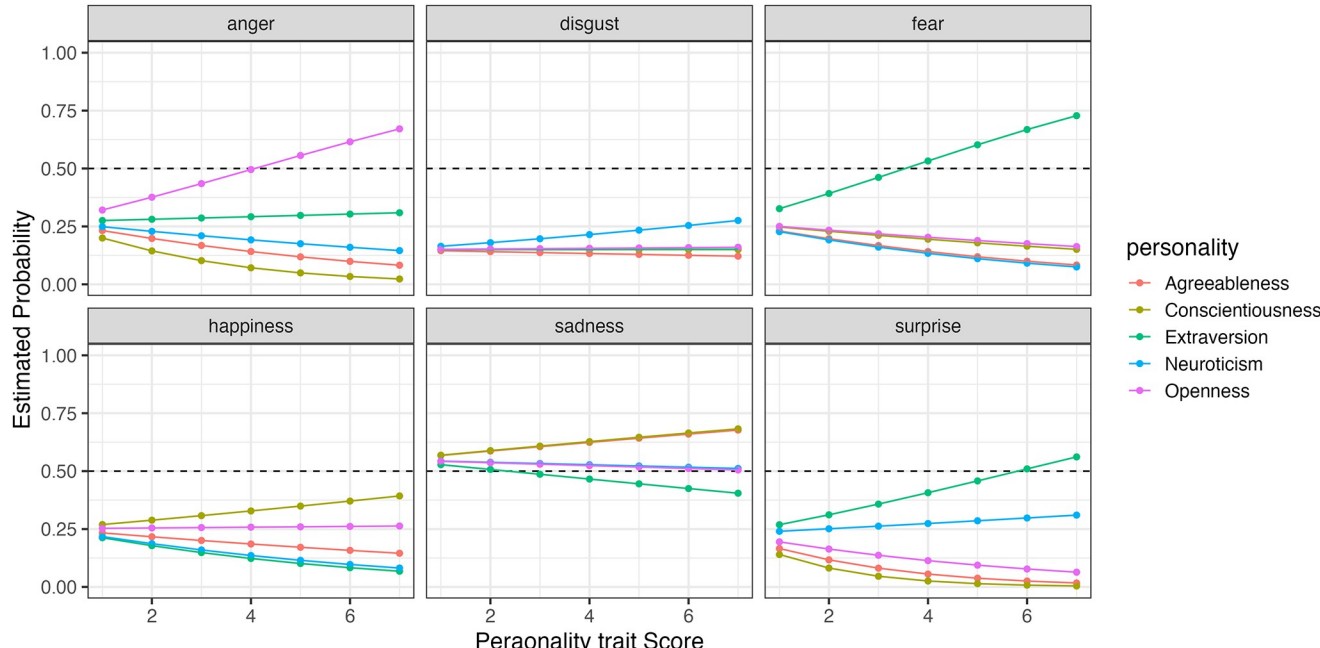

**Fig 7. The probability of selecting each expression as the five personality traits changed when faces expressing sadness were presented in Experiment Part 1.**

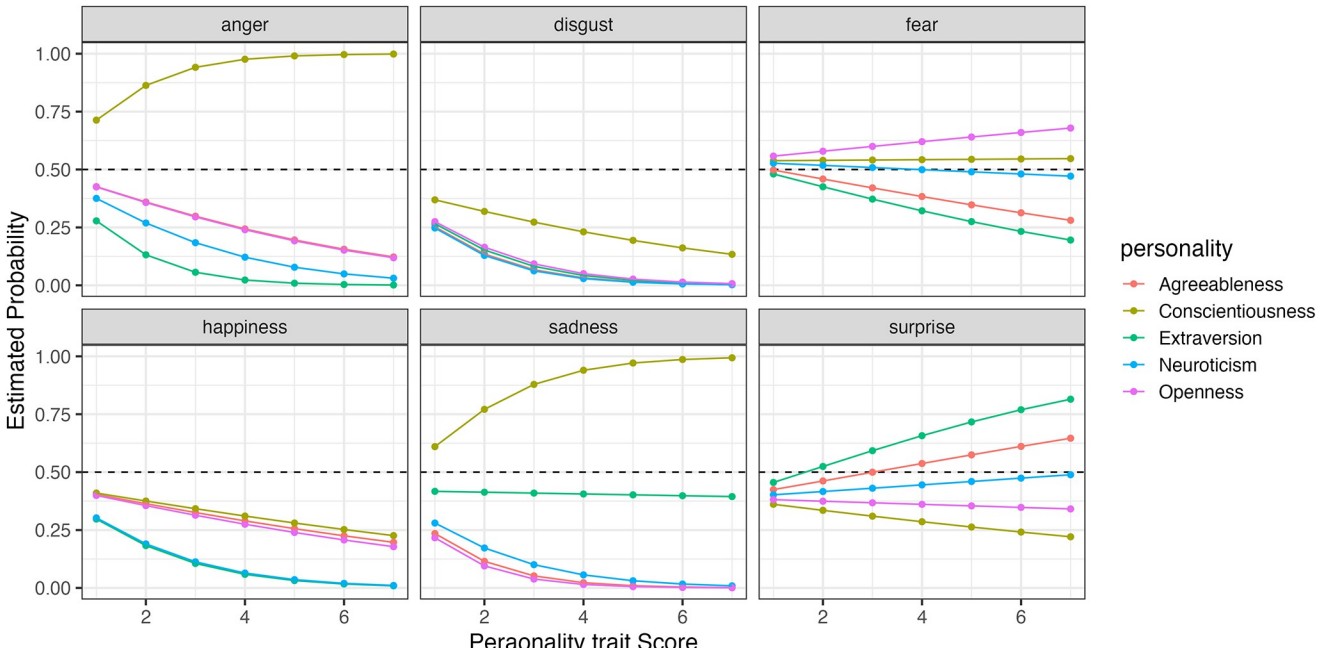

**Fig 8. The probability of selecting each expression as the five personality traits changed when faces expressing surprise were presented in Experiment Part 1.**

scores of the other four traits were fixed at zero. As shown in Fig 6, there was a tendency for individuals with high levels of Agreeableness (red) to choose happiness but not to choose anger, disgust and sadness. Fig 7 shows that individuals with low levels of Conscientiousness (yellow) tended to choose anger. Fig 8 shows that individuals with high levels of Extraversion

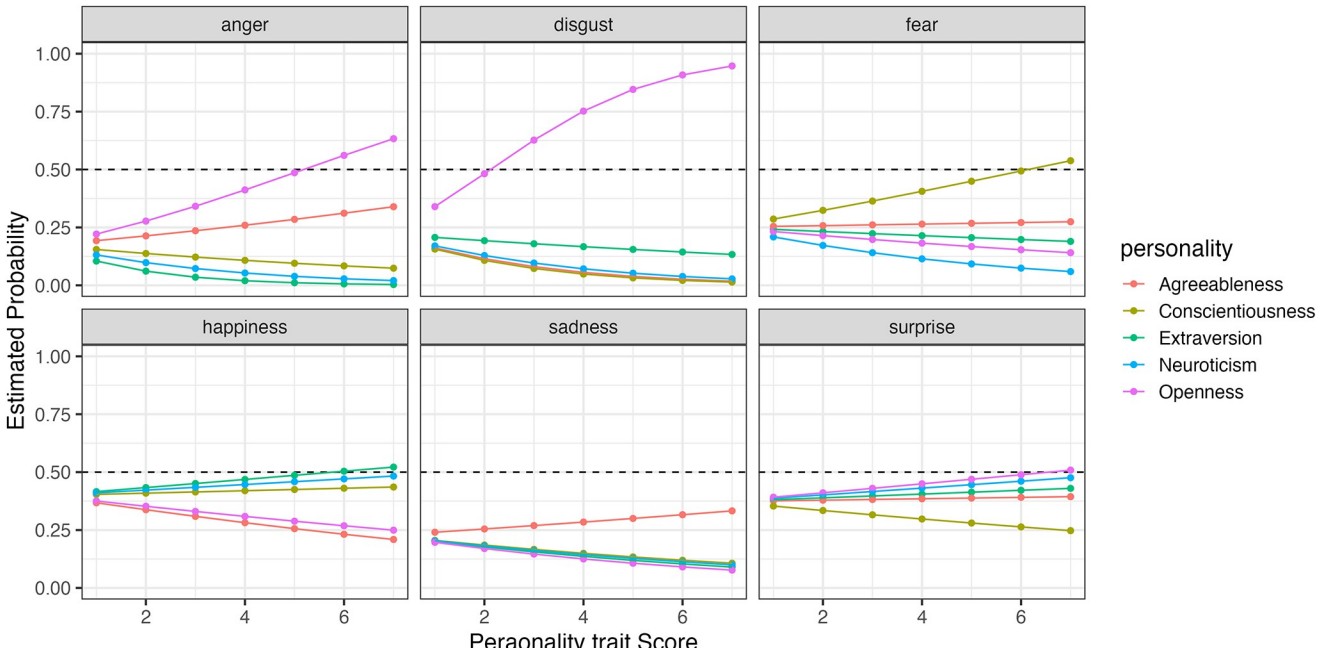

**Fig 9. The probability of selecting each expression as the five personality traits changed when faces expressing fear were presented in Experiment Part 1.**

(green) tended to choose surprise but not to choose anger and fear. Fig 9 shows that individuals with high levels of Openness (pink) tended to choose disgust and anger. Conversely, individuals with high levels of Conscientiousness (yellow) tended to choose fear and not to choose surprise. These results suggest that when expressions are expressed ambiguously (e.g. landmarks), observers may refer to their underlying characteristics (personality traits) when making choices.

### Result 2

Part 2 of the present study examined the relationship between personality traits, observational behavior, and recognition of facial expressions while imposing observation restrictions on participants. Fig 10 shows the estimated recognition for each expression stimulus by the model, with the dashed line indicating the chance level. The results were similar to those of Part 1. There were tendencies to correctly recognize positive expressions (happiness and surprise), while recognition rates were lower for negative expressions. Specifically, the estimated recognition rates for happiness for surprise were comparable to Part 1. Moreover, a high tendency for misidentifying fear as surprise was observed, consistent with Part 1. Moreover, a high tendency for misidentifying fear as surprise was observed, consistent with Part 1.

Table 2 summarizes the significant relationships between the participants' personality traits and their selections about facial expressions. The results show that people with high Agreeableness tended to correctly recognize happiness and surprise. People with high levels of Extraversion tended to correctly recognize surprise. On the other hand, people with high in Conscientiousness and Extraversion tended to misidentify sadness as happiness and anger, respectively, and people with high Neuroticism tended to misidentify fear as surprise. Additionally, people with high in Extraversion tended to not recognize sadness as sadness, and people with high in Neuroticism tended to not recognize fear as fear. Compared to the results from

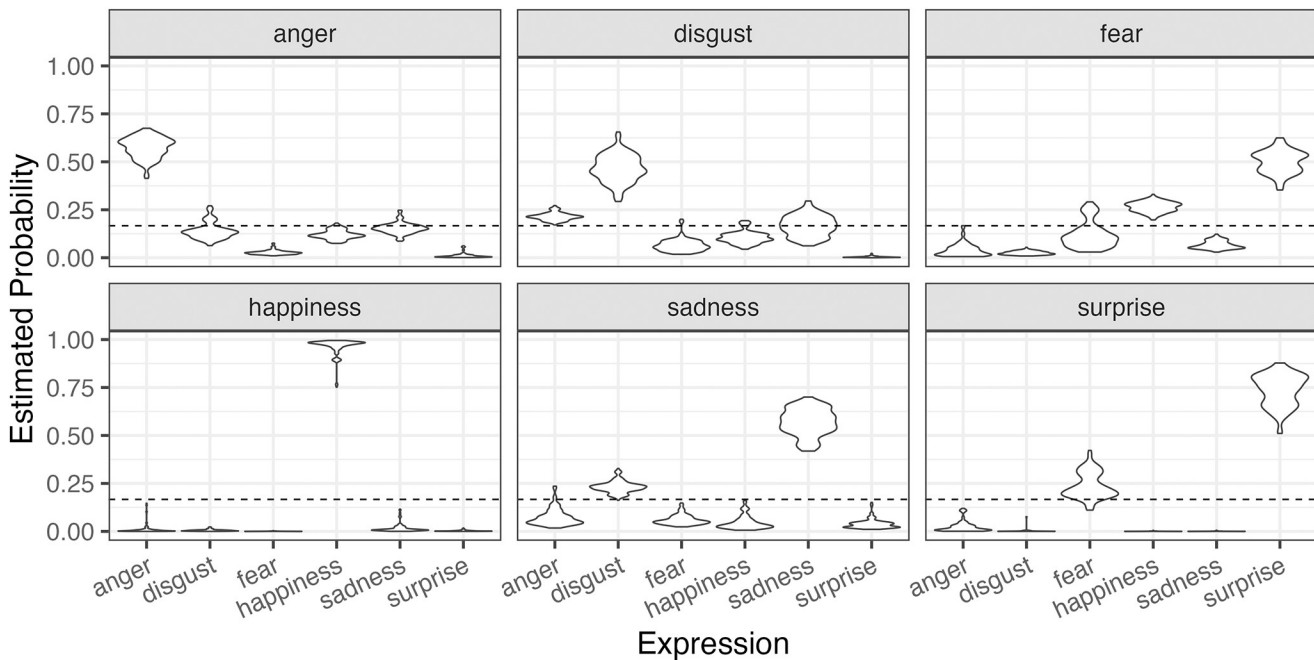

**Fig 10. Predictive posterior distributions of selecting each expression type for each condition (emotional expression) in Experiment Part 2.**

**Table 2. Significant personality traits for selecting and or not selecting each expression type for each condition in Experiment Part 2.**

| Expression | Response | Predictor | Mean | 95%HDI |
|---|---|---|---|---|
| Happiness | Happiness | Agree. | 0.812 | 0.395 ⌣ 1.321 |
| | Anger | Agree. | -1.472 | -2.375 ⌣ -0.634 |
| | Sadness | Agree. | -0.736 | -1.376 ⌣ -0.197 |
| Sadness | Happiness | Consc. | 0.385 | 0.004 ⌣ 0.771 |
| | | Neuro. | -0.353 | -0.693 ⌣ -0.025 |
| | Anger | Extra. | 0.346 | 0.044 ⌣ 0.670 |
| | Sadness | Extra. | -0.184 | -0.346 ⌣ -0.004 |
| | Surprise | Neuro. | -0.410 | -0.774 ⌣ -0.094 |
| Surprise | Surprise | Agree. | 0.218 | 0.001 ⌣ 0.443 |
| | | Extra. | 0.266 | 0.059 ⌣ 0.465 |
| | Anger | Extra. | -0.881 | -1.547 ⌣ -0.271 |
| | Fear | Agree. | -0.255 | -0.484 ⌣ -0.031 |
| Anger | Surprise | Neuro. | -0.701 | -1.320 ⌣ -0.084 |
| Disgust | Sadness | Consc. | -0.304 | -0.552 ⌣ -0.045 |
| Fear | Surprise | Neuro. | 0.171 | 0.013 ⌣ 0.344 |
| | Anger | Extra. | -0.620 | -1.089 ⌣ -0.179 |
| | Fear | Neuro. | -0.339 | -0.567 ⌣ -0.101 |

Part 1 (i.e., the same task without restrictions), the tendency for people with high Agreeableness to correctly recognize positive expressions (happiness and surprise) slightly increased, and the tendency not to select negative expressions for positive ones has been confirmed. For example, happiness was less likely to recognized as sadness or anger and not recognizing surprise as fear. However, people with high Extraversion, in general, tended not to select negative expressions for both positives and negatives, for example, not selecting sadness when observing sadness faces. One possible reason for these results is that in Part 2, where observational behavior was restricted, resulting in conscious observational behaviors leading to such tendencies.

To compare how changes in the five personality traits affect the recognition rates for each expression, we simulated the results using the model. Fig 11 shows the simulated selection probabilities for each expression with changes in the five personality traits when faces expressing happiness were presented (Figs 12–14 for sadness, surprise and fear, respectively).

When happiness faces were presented (Figs 6 for Part I and 11 for Part 2), the changes in selection rates associated with changes in Agreeableness were similar in Part 1 and Part 2. When fear faces were presented, there was a significant reduction in the misidentification of fears as disgust for individuals with high levels of Openness (pink) but almost no change for anger (disgust and anger planes in Figs 9 and 14). Comparing these results of Parts 1 and 2, the general trend in selection rates, when fear faces were presented, did not show much difference, except that for individuals with high levels of Openness. Specifically, the "big mistake" of high Openness individuals recognizing fear as disgust in Part 1 disappeared, possibly due to the effect of restricting them to have only conscious observational behavior, reducing the misidentification rate.

One possible reason for somewhat different results about the relationships between personality traits and selections of expressions between Part 1 and Part 2 was the influence of observational behaviors (i.e., unrestricted or restricted). Therefore, we summarized the relationship between observational behavior and selections of expressions in Table 3. There was a tendency to accurately recognize fear when viewing the middle part of the face. When the upper (lower) part of the face was more likely to be seen, the recognition rates of happiness and anger

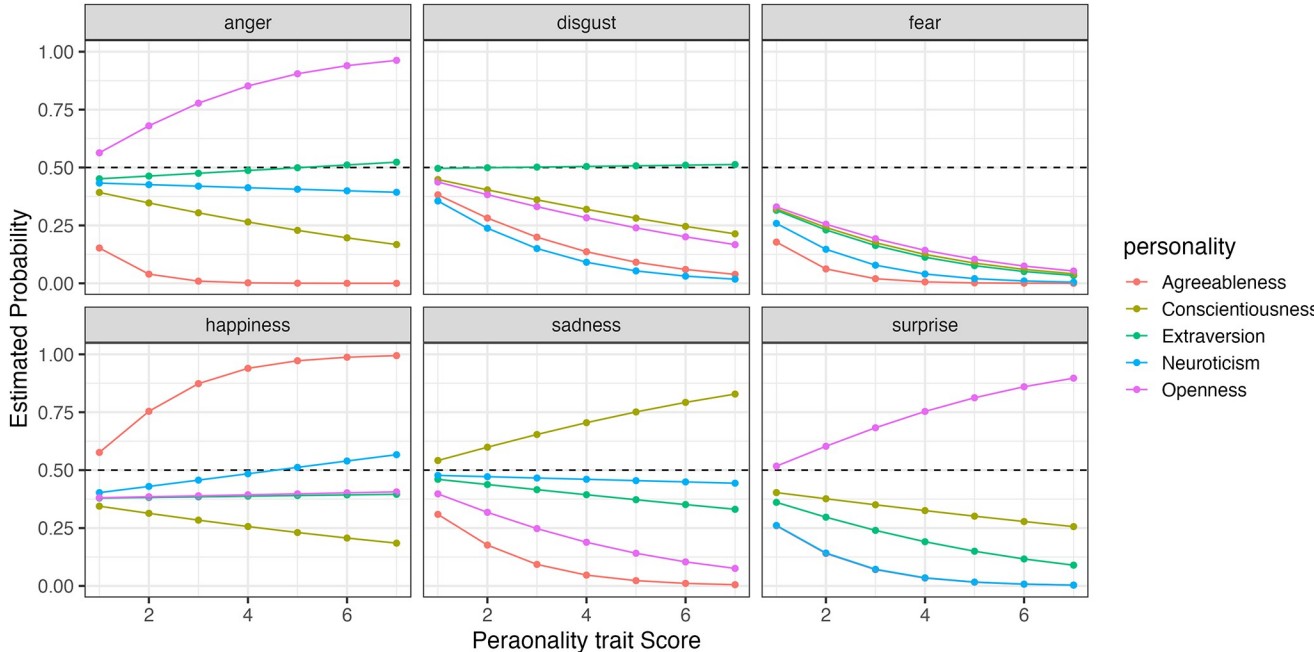

**Fig 11. The probability of selecting each expression as the five personality traits changed when faces expressing happiness were presented in Experiment Part 2.**

(happiness) were low. In particular, when the lower part of the face was more likely to be seen, there was a high tendency to mistakenly recognize happiness as sadness or disgust, while when the upper part was seen, there was a high tendency to mistakenly recognize fear as surprise. The results of the present and previous analyses show an interesting phenomenon. While

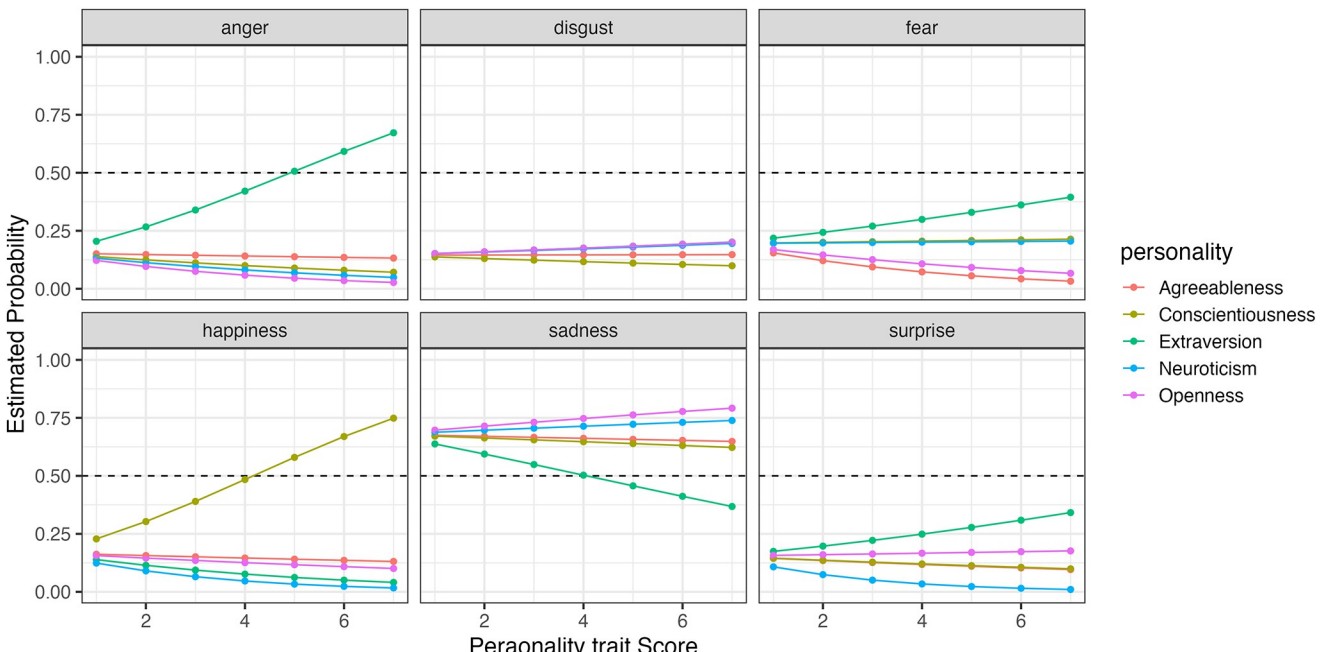

**Fig 12. The probability of selecting each expression as the five personality traits changed when faces expressing sadness were presented in Experiment Part 2.**

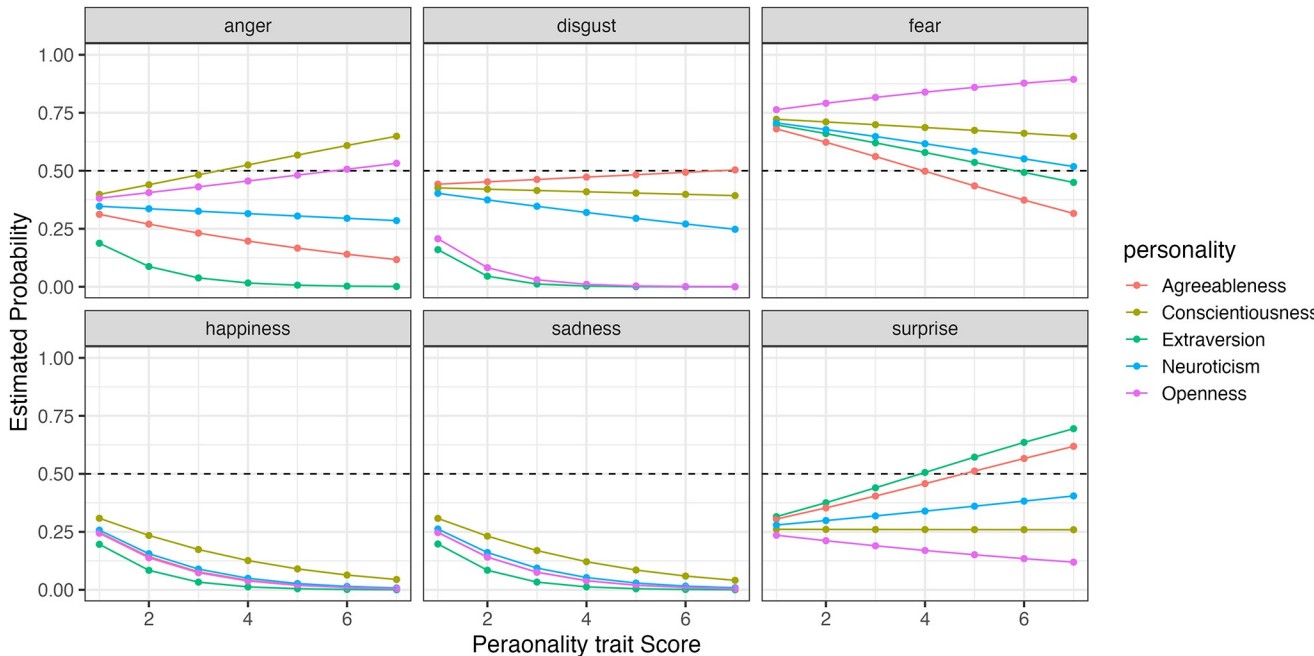

**Fig 13. The probability of selecting each expression as the five personality traits changed when faces expressing surprise were presented in Experiment Part 2.**

differences in personality traits made recognition of positive expressions easier, differences in observational behaviors made recognition of negatives. In order to investigate the cause of such apparent contradictions, it is necessary to examine the relationship between personality traits and observational behaviors in facial recognition tasks.

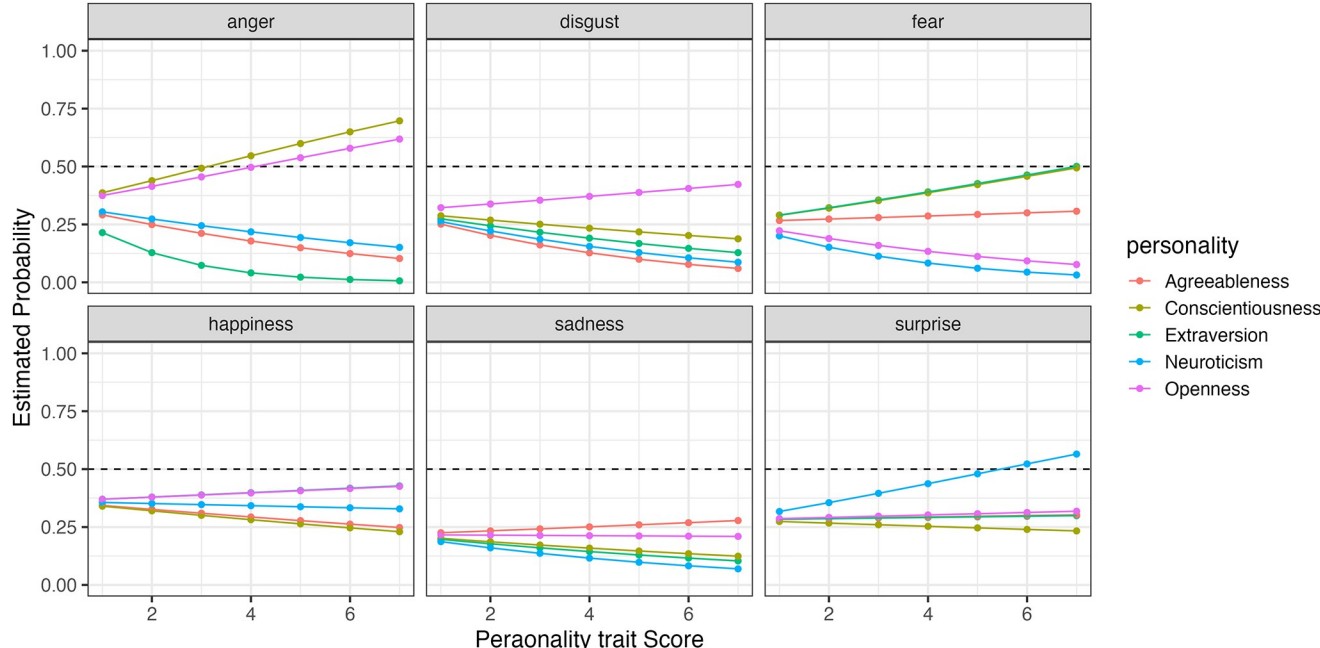

**Fig 14. The probability of selecting each expression as the five personality traits changed when faces expressing fear were presented in Experiment Part 2.**

**Table 3. Significant observation behaviors for selecting and or not selecting each expression type for each condition in Experiment Part 1.**

| Expression | Response | Area | Mean | 95%HDI |
|---|---|---|---|---|
| Happiness | Happiness | Upper | -767.144 | -1238.152 ⌣ -289.434 |
| | | Low | -1016.051 | -1650.876 ⌣ -481.775 |
| | Disgust | Low | 856.070 | 61.954 ⌣ 1810.830 |
| | Sadness | Low | 1098.640 | 484.150 ⌣ 1765.882 |
| Fear | Fear | Middle | 161.977 | 12.126 ⌣ 320.643 |
| | Surprise | Upper | 164.602 | 12.879 ⌣ 315.883 |
| | Happiness | Middle | -258.489 | -427.295 ⌣ -104.806 |
| Anger | Anger | Upper | -230.716 | -391.424 ⌣ -76.173 |
| | Happiness | Middle | -280.260 | -508.078 ⌣ -10.772 |

To examine the results that suggest the participants' observational behavior and personality traits have contradictory effects on the recognition of facial expressions, we analyzed the relationship between personality traits and observational behavior using the method described in Analysis C. Table 4 shows the summary of the result. The Bernoulli distribution represents whether or not a particular part of faces was observed at least once, and the beta distribution represents how much. The results can be summarized as follows:

When faces expressing happiness were presented, people with high Agreeableness and Conscientiousness tended to look at the lower part of the faces at least once, whereas those with high Extraversion tended not to look at them at all. People with high Openness tended not to look at the lower part of the faces.

When faces expressing anger were presented, people with high Extraversion tended not to look at the middle part of the face.

**Table 4. Significant predictors in ZIB models in Experiment Part 2.**

| Model | Expressoin | Area | Predictor | Mean | 95%HDI |
|---|---|---|---|---|---|
| Bernoulli | Happiness | Low | Agree. | 0.573 | 0.074 ⌣ 1.058 |
| | | | Consc. | 0.406 | 0.004 ⌣ 0.796 |
| | | | Extra. | -0.599 | -0.993 ⌣ -0.212 |
| | Disgust | Upper | Neuro. | 112.397 | 29.346 ⌣ 197.044 |
| | | | Open. | 94.811 | 14.700 ⌣ 164.198 |
| | | | Agree. | -63.562 | -188.107 ⌣ -16.042 |
| | | | Consc. | -74.106 | -140.940 ⌣ -15.275 |
| | | Low | Extra. | -0.476 | -0.950 ⌣ -0.028 |
| | Sadness | Low | Extra. | -0.445 | -0.847 ⌣ -0.084 |
| | Fear | Low | Extra. | -0.585 | -1.063 ⌣ -0.121 |
| Beta | Happiness | Low | Open. | -0.206 | -0.392 ⌣ -0.037 |
| | Anger | Middle | Extra. | -0.127 | -0.224 ⌣ -0.030 |
| | Disgust | Upper | Extra. | -0.099 | -0.182 ⌣ -0.015 |
| | | Middle | Extra. | -0.124 | -0.228 ⌣ -0.019 |
| | | Low | Open. | -0.161 | -0.317 ⌣ -0.003 |
| | Surprise | Low | Agree. | 0.241 | 0.048 ⌣ 0.457 |
| | | | Extra. | -0.156 | -0.322 ⌣ -0.006 |
| | | | Open. | -0.192 | -0.345 ⌣ -0.032 |
| | Fear | Middle | Extra. | -0.132 | -0.238 ⌣ -0.028 |

When faces expressing disgust were presented, people with high Neuroticism and Openness tended to look at the upper part of the face at least once, whereas those with high Conscientiousness and Agreeableness tended not to look at the upper part at all. Those with high Extraversion tended not to look at the lower part of the face at all. Those with high Extraversion tended not to look at the upper and middle parts of the faces, and those with high Openness tended not to look at the lower part.

When faces expressing surprise were presented, people with high Agreeableness tended to look at the lower part of the face for a long time, whereas those with high Extraversion and Openness tended not to.

When faces expressing sadness were presented, people with high Extraversion tended not to look at the lower part of the face at all.

## Discussion and conclusion

The study had two main objectives. The first was to investigate whether humans could correctly recognize facial expressions when facial features were represented by a limited number of landmarks, a widely used approach in expression recognition in machine learning. The second was to examine whether there were differences in conscious observational behavior when recognizing different facial expressions. To achieve these objectives, we conducted an experiment that consisted of two parts, each having its own objectives. In Part 1, participants could freely observe video images of facial expressions represented by landmarks and judged the expressions. In Part 2, participants performed the same expression recognition task, but they could only observe the region where the mouse cursor was located, having limited information on the faces. The responses to the facial expression recognition task in Part 1 were used as a basis and compared to those of Part 2. The active observational behavior forced by limited view areas in Part 2 enabled us to measure what we called "conscious observational behavior" for specific facial expressions.

The results of Part 1 showed that humans could correctly recognize facial expressions even when the faces were represented by a small set of landmarks. The results of Part 1 and Part 2 showed great similarity, except for judging faces expressing fear. The accuracy of recognizing faces expressing happiness exceeded 95% (the mean of the posterior distribution). Conversely, a tendency to misidentify fear as surprise was also observed. These results were consistent with previous studies on machine learning [13]. That is, even with a minimal amount of facial expression information, humans can recognize positive expressions, but negative expressions are still difficult to recognize. However, our results were somewhat higher than those on machine learning. Our results showed several significant correlations between personality traits, observational behaviors, and recognition accuracies. For example, people with high Agreeableness did not consciously look at the upper and lower areas of the face when judging happiness, which in turn caused them to recognize faces expressing happiness accurately. Also, people with high Neuroticism did not look at the middle of the face for long enough when judging fear, which in turn caused them not to recognize faces expressing fear accurately. When we examined the stimulus videos used in the experiment, we found that faces expressing happiness, surprise, and fear tended to have mouth-opening movements. If participants focused excessively on the lower area of the face, they might be misled by mouth-opening movement and then result in misrecognition. In fact, when faces expressing fear were presented, they were often misrecognized as surprise or happiness. From these results, the participants' personality traits led to conscious observational behaviors that obtained information used for facial expression judgments. These results are consistent with those of previous studies [14, 27]. In other words, personality traits and observational behavior are strongly

correlated, and the information obtained through observational behavior also influences people's cognition and judgment formation. This effect is also present in conscious observational behavior. Additionally, the higher rate of errors in Part 1 compared to Part 2 suggests that information obtained through unconscious observational behaviors may have interfered with accurate recognitions or forced conscious observational behaviors may have promoted better recognitions.

In this experiment, we used videos of facial features represented by landmarks as stimuli to elicit facial expressions. Despite the fact that these expressions were unfamiliar to humans and were expected to result in inaccurate recognitions, some positive expressions were accurately recognized. Nevertheless, there was also a noticeable tendency to not choose negative expressions. There are some issues that need to be addressed. While this study focused on investigating observer characteristics, the influence of the stimuli cannot be ignored. Additionally, although the study confirmed the focus of attention for each expression, we did not confirm the priority of each feature, as we did not set a limit on the observation time nor we did not treated our data as time-series data. Using dynamics of observational behaviors could contribute not only to the elucidation of the mechanism of human facial recognition but also to improving the accuracy of facial recognition in machine learning. In addition, the size of the visual range may also influence judgments. For example, a smaller scope may limit the amount of information that can be observed at a time, which may lead to lower accuracy (due to less information) or higher accuracy (due to concentrated attention).

## Acknowledgments

I am truly grateful for the tremendous support from Professor Hiroki Shinkawa for the present research.

## Author Contributions

**Conceptualization:** Kuangzhe Xu.

**Data curation:** Kuangzhe Xu.

**Formal analysis:** Kuangzhe Xu.

**Methodology:** Kuangzhe Xu, Toshihiko Matsuka.

**Supervision:** Toshihiko Matsuka.

**Validation:** Kuangzhe Xu, Toshihiko Matsuka.

**Visualization:** Kuangzhe Xu.

**Writing – original draft:** Kuangzhe Xu.

**Writing – review & editing:** Kuangzhe Xu, Toshihiko Matsuka.

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
