## [Decision Letter · Decision Letter 0]

24 Jul 2023

PONE-D-23-10601Conscious observational behavior in recognizing landmarks in facial expressionsPLOS ONE

Dear Dr. xu,

Thank you for submitting your manuscript to PLOS ONE. After careful consideration, we feel that it has merit but does not fully meet PLOS ONE’s publication criteria as it currently stands. Therefore, we invite you to submit a revised version of the manuscript that addresses the points raised during the review process.

 Please submit your revised manuscript by Sep 07 2023 11:59PM. If you will need more time than this to complete your revisions, please reply to this message or contact the journal office at plosone@plos.org. Please include the following items when submitting your revised manuscript:A rebuttal letter that responds to each point raised by the academic editor and reviewer(s). You should upload this letter as a separate file labeled 'Response to Reviewers'.A marked-up copy of your manuscript that highlights changes made to the original version. You should upload this as a separate file labeled 'Revised Manuscript with Track Changes'.An unmarked version of your revised paper without tracked changes. You should upload this as a separate file labeled 'Manuscript'.If applicable, we recommend that you deposit your laboratory protocols in protocols.io to enhance the reproducibility of your results. Protocols.io assigns your protocol its own identifier (DOI) so that it can be cited independently in the future. For instructions see: https://journals.plos.org/plosone/s/submission-guidelines#loc-laboratory-protocols. Additionally, PLOS ONE offers an option for publishing peer-reviewed Lab Protocol articles, which describe protocols hosted on protocols.io. Read more information on sharing protocols at https://plos.org/protocols?utm_medium=editorial-email&utm_source=authorletters&utm_campaign=protocols.

We look forward to receiving your revised manuscript.

Kind regards,

Vijayalakshmi G V Mahesh, Ph.D

Academic Editor

PLOS ONE

Journal Requirements:

Reviewers' comments:

Reviewer's Responses to Questions

**Comments to the Author**

1. Is the manuscript technically sound, and do the data support the conclusions?

Reviewer #1: Yes

Reviewer #2: Yes

2. Has the statistical analysis been performed appropriately and rigorously? 

Reviewer #1: Yes

Reviewer #2: Yes

3. Have the authors made all data underlying the findings in their manuscript fully available?

Reviewer #1: Yes

Reviewer #2: Yes

4. Is the manuscript presented in an intelligible fashion and written in standard English?

Reviewer #1: Yes

Reviewer #2: Yes

5. Review Comments to the Author

Reviewer #1: The manuscript is described in a technically sound way. The scientific research with data that supports the conclusions is provided. Experiments have been conducted which is described in a elaborate way which is also provided with appropriate results. Care has to be taken as there are repetition of words and sentences.

Reviewer #2: Constructed Bayesian models to analyze influence of personality traits and observational behaviors on expressional judgments. The objective of the work is met to reasonable extent.

1) what threshold of the parameter is deciding upon the minimum or maximum facial expressions in situations.

2) "The existence of a mechanism whereby personality traits lead to different conscious observational behaviors, and expressional judgments are based on information obtained through those observational behaviors." , little elaboration of the Mechanism is needed

3) After deleting all identifiable information of the 79 participants and are storing the data, what is the actual data information size. Any particular reason for considering the data where female are 74 and male are 12 as participants. what is the impact of this uneven consideration of the data in terms of gender to the research carried.

4)few more comparison on choosing the Bayesian model for the analysis.

further the six expressions along with personality traits are graphed well. More insight on the assessment on personality traits can be added.

5) More information on personality trait score can be added.

6) Overall good analysis... more information to be added on the impact of adding more data than what is considered. how will the Bayesian model handle its efficiency with more number of participants.

6. PLOS authors have the option to publish the peer review history of their article (what does this mean?). If published, this will include your full peer review and any attached files.

Reviewer #1: No

Reviewer #2: **Yes: **ROOPA B S

---

## [Author Response · Author response to Decision Letter 0]

25 Aug 2023

Reviewer #1: The manuscript is described in a technically sound way. The scientific research with data that supports the conclusions is provided. Experiments have been conducted which is described in a elaborate way which is also provided with appropriate results. Care has to be taken as there are repetition of words and sentences.

>Thank you very much for your valuable suggestion. We re-read our paper and realized that there were several sentences that were not needed or ambiguous. We have double-checked the content and corrected instances of word and sentence repetition to make our paper better.

Reviewer #2: Constructed Bayesian models to analyze influence of personality traits and observational behaviors on expressional judgments. The objective of the work is met to reasonable extent.

1) what threshold of the parameter is deciding upon the minimum or maximum facial expressions in situations.

>[Line 119 - 120] We have added content about the criteria for determining the visible range. The size of the visible range was determined based on the content of the previous studies [line 136 – 138].

[Line 406 - 409 ] We have added a discussion of the effect of visible range size.

2) "The existence of a mechanism whereby personality traits lead to different conscious observational behaviors, and expressional judgments are based on information obtained through those observational behaviors." , little elaboration of the Mechanism is needed

>[line 391 - 394] we add a few sentences to elaborate and discuss the Mechanism. 

3) After deleting all identifiable information of the 79 participants and are storing the data, what is the actual data information size. Any particular reason for considering the data where female are 74 and male are 12 as participants. what is the impact of this uneven consideration of the data in terms of gender to the research carried.

>[line 73 - 81] our participant sample size was 41(12 male and 29 female). As you pointed out, the ratio of male to female participants is somewhat unbalanced. However, since we did not examine the effect of gender in this experiment, we do not believe that this imbalance would affect our main results.　 Our data showed a significant correlation between gender and personality traits, and using both variables simultaneously in the analysis may lead to multicollinearity. Thus, we left the “gender effect’ from our analyses. In the future, we would like to consider the gender effect in our research. 

4)few more comparison on choosing the Bayesian model for the analysis.

further the six expressions along with personality traits are graphed well. More insight on the assessment on personality traits can be added.

>[Line 251 - 266] we add other results about happiness, sadness and surprise.

[Line 296 - 310] we add other results about happiness.

5) More information on personality trait score can be added.

>[Line 124 - 128] we add new content to explain the TIPI-J and Big Five (by using a footnote). 

6) Overall good analysis... more information to be added on the impact of adding more data than what is considered. how will the Bayesian model handle its efficiency with more number of participants.

>[Line 157 -164] we add new content to talk about the Bayesian model.

---

## [Decision Letter · Decision Letter 1]

5 Sep 2023

Conscious observational behavior in recognizing landmarks in facial expressions

PONE-D-23-10601R1

Dear Dr. xu,

We’re pleased to inform you that your manuscript has been judged scientifically suitable for publication and will be formally accepted for publication once it meets all outstanding technical requirements.

Kind regards,

Vijayalakshmi G V Mahesh, Ph.D

Academic Editor

PLOS ONE

Additional Editor Comments (optional):

Reviewers' comments:

Reviewer's Responses to Questions

**Comments to the Author**

1. If the authors have adequately addressed your comments raised in a previous round of review and you feel that this manuscript is now acceptable for publication, you may indicate that here to bypass the “Comments to the Author” section, enter your conflict of interest statement in the “Confidential to Editor” section, and submit your "Accept" recommendation.

Reviewer #1: All comments have been addressed

Reviewer #2: All comments have been addressed

2. Is the manuscript technically sound, and do the data support the conclusions?

Reviewer #1: Yes

Reviewer #2: Yes

3. Has the statistical analysis been performed appropriately and rigorously? 

Reviewer #1: Yes

Reviewer #2: Yes

4. Have the authors made all data underlying the findings in their manuscript fully available?

Reviewer #1: Yes

Reviewer #2: Yes

5. Is the manuscript presented in an intelligible fashion and written in standard English?

Reviewer #1: Yes

Reviewer #2: Yes

6. Review Comments to the Author

Reviewer #1: (No Response)

Reviewer #2: Queries raised are addressed appropriately.

They have added content about the criteria for determining the visible range. The size of the visible range was determined based on the content of the previous studies.

for the question :Any particular reason for considering the data where female are 74 and male are 12 as participants. what is the impact of this uneven consideration of the data in terms of gender to the research carried.

>[line 73 - 81] our participant sample size was 41(12 male and 29 female). As pointed out, the ratio of male to female participants is somewhat unbalanced. However, since they did not examine the effect of gender in this experiment, they do not believe that this imbalance would affect their main results. Their data showed a significant

correlation between gender and personality traits, and using both variables simultaneously in the analysis may lead to multicollinearity. Thus, they left the “gender effect’ from their analyses. In the future, they would like to consider the gender effect in their research.

I recommend the work done to be accepted.

7. PLOS authors have the option to publish the peer review history of their article (what does this mean?). If published, this will include your full peer review and any attached files.

Reviewer #1: No

Reviewer #2: **Yes: **ROOPA B S

---

## [Editor Report · Acceptance letter]

12 Sep 2023

PONE-D-23-10601R1 

Conscious observational behavior in recognizing landmarks in facial expressions 

Dear Dr. Xu:

I'm pleased to inform you that your manuscript has been deemed suitable for publication in PLOS ONE. Congratulations! Your manuscript is now with our production department. 

Kind regards, 

on behalf of

Dr. Vijayalakshmi G V Mahesh 

Academic Editor

PLOS ONE